# Influence of Individual Bracket Base Design on the Shear Bond Strength of In-Office 3D Printed Brackets—An In Vitro Study

**DOI:** 10.3390/jfb14060289

**Published:** 2023-05-24

**Authors:** Lutz D. Hodecker, Mats Scheurer, Sven Scharf, Christoph J. Roser, Ahmed M. Fouda, Christoph Bourauel, Christopher J. Lux, Carolien A. J. Bauer

**Affiliations:** 1Department of Orthodontics and Dentofacial Orthopedics, University of Heidelberg, Im Neuenheimer Feld 400, 69120 Heidelberg, Germany; lutz.hodecker@med.uni-heidelberg.de (L.D.H.); christoph.roser@med.uni-heidelberg.de (C.J.R.); 2Department of Oral and Maxillofacial Surgery, University of Heidelberg, Im Neuenheimer Feld 400, 69120 Heidelberg, Germany; mats.scheurer@med.uni-heidelberg.de; 3Private Practice of Orthodontics, 50933 Cologne, Germany; svenscharf@me.com; 4Oral Technology, Dental School, Medical Faculty, University Hospital of Bonn, Welschnonnenstr 17, 53111 Bonn, Germany; ah.fouda@gmail.com (A.M.F.); bourauel@uni-bonn.de (C.B.)

**Keywords:** 3D-printed, individual braces, shear bond strength, polymer, customized braces, bracket design

## Abstract

(1) Background: Novel high-performance polymers for medical 3D printing enable in-office manufacturing of fully customized brackets. Previous studies have investigated clinically relevant parameters such as manufacturing precision, torque transmission, and fracture stability. The aim of this study is to evaluate different design options of the bracket base concerning the adhesive bond between the bracket and tooth, measured as the shear bond strength (SBS) and maximum force (F_max_) according to DIN 13990. (2) Methods: Three different designs for printed bracket bases were compared with a conventional metal bracket (C). The following configurations were chosen for the base design: Matching of the base to the anatomy of the tooth surface, size of the cross-sectional area corresponding to the control group (C), and a micro- (A) and macro- (B) retentive design of the base surface. In addition, a group with a micro-retentive base (D) matched to the tooth surface and an increased size was studied. The groups were analyzed for SBS, F_max_, and adhesive remnant index (ARI). The Kruskal–Wallis test with a post hoc test (Dunn–Bonferroni) and Mann–Whitney U test were used for statistical analysis (significance level: *p* < 0.05). (3) Results: The values for SBS and F_max_ were highest in C (SBS: 12.0 ± 3.8 MPa; F_max_: 115.7 ± 36.6 N). For the printed brackets, there were significant differences between A and B (A: SBS 8.8 ± 2.3 MPa, F_max_ 84.7 ± 21.8 N; B: SBS 12.0 ± 2.1 MPa, F_max_ 106.5 ± 20.7 N). F_max_ was significantly different for A and D (D: F_max_ 118.5 ± 22.8 N). The ARI score was highest for A and lowest for C. (4) Conclusions: This study shows that conventional brackets form a more stable bond with the tooth than the 3D-printed brackets. However, for successful clinical use, the shear bond strength of the printed brackets can be increased with a macro-retentive design and/or enlargement of the base.

## 1. Introduction

Innovative approaches demonstrate modern computer-aided design and computer-aided manufacturing (CAD/CAM) solutions for in-office manufacturing of fully customizable bracket systems made of high-performance polymers. The potential advantages of these approaches are seen in the individual color design, the individual programming of bracket parameters, the modeling of the bracket and positioning aid in combination, the adaptation of bracket geometry for hybrid appliances, and the increase in added value for the orthodontic practice. Prior to the general clinical use of such CAD/CAM solutions, material science studies based on orthodontic bracket standards can ensure patient safety, provide a qualitative comparison with existing bracket systems, and contribute to the brackets’ design in the practice. Bond strength is one of the most important aspects for the clinical performance of an orthodontic bracket and should be tested prior to bringing it onto the market, best in a standardized setup.

Various esthetically optimized bracket systems are available in fixed orthodontic therapy. The lingual bracket systems are considered particularly efficient and esthetic [1]. Vestibular systems are still considered the gold standard due to their wide use and ease of handling. For vestibular systems, ceramic brackets are preferred to polymer brackets as an esthetic solution, although polymer brackets cause significantly less friction in the bracket–arch complex [2,3,4]. The main disadvantages of polymer brackets are their tendency to discoloration and lack of mechanical stability [5,6,7,8]. In addition, the adhesion of polymer brackets to the tooth surface, measured as shear bond strength, is poorer than that of metal and ceramic brackets and may result in less efficient treatment [9,10,11,12]. On the one hand, the shear bond strength should not exceed a value that leads to the tearing of the enamel when the brackets are removed, and on the other hand, the shear bond strength should not fall below a value that leads to the loss of the bracket during therapy [13,14]. The range of ideal shear bond strength should be between 5.9 MPa and 7.9 MPa, according to Reynolds, or between 5 MPa and 10 MPa, according to Diedrich [13,15].

A high bracket loss rate is considered particularly detrimental in clinical practice, as it prolongs the duration of therapy and increases the number of unscheduled appointments [16]. Therefore, the measurement of shear bond strength is considered an essential tool in evaluating and selecting a suitable bracket system. Shear bond strength depends on the bracket materials, adhesives, tooth surface, and base morphology [16,17,18,19]. Factors affecting the bracket base morphology include size, retention pattern, surface treatment, and shape, which can occur during manufacturing and customization [20,21]. In addition to the measurement of shear bond strength, observation of the fracture location of the adhesion in the shear test can also provide information about the adhesive bond between the tooth surface and the bracket base. The ARI score was defined by Artun and Bergland in 1984 to evaluate the breaking point of the adhesive bond (Table 1) [22]. The classification is based on the location of the fracture. It may be located between the bracket and the adhesive, between the adhesive and the enamel, or at both interfaces.

It was the aim of the presented in vitro study to evaluate different design options for the bracket base of in-office manufactured bracket systems concerning the adhesive bond between the bracket base and the tooth surface using an established test method according to DIN 13990 [23]. A commercially available bracket that has been studied extensively in earlier studies should serve as a gold standard.

**Null hypothesis**. The adhesive bond between differently configured printed bracket bases and the surface of bovine teeth is weaker than that of conventional metal brackets and is clinically insufficient. The retentive design and the size of the cross-sectional area of the printed brackets have no significant influence on the bond between the printed brackets and the tooth surface.

## 2. Materials and Methods

Shear bond strength testing was performed on erupted permanent bovine mandibular incisors according to the requirements of DIN standards 13990 and 3696 [23,24]. Ten test specimens were tested per bracket group (*N* = 10). In the first test procedure, two different 3D-printed bracket systems with a micro-retentive (group A) and a macro-retentive (group B) design of the base were compared with a control group C consisting of a conventional metallic bracket system (discovery^®^, Dentaurum GmbH & Co. KG, Ispringen, Germany). The bases of the tested groups had the same cross-sectional area and were, therefore, the same size. Subsequently, in a second test procedure, the influence of increasing the size of the bracket base for the 3D-printed brackets was investigated regarding shear bond strength.

The printed brackets were previously manufactured using the stereolithography printing process (SLA) with a Formlabs 3b printer (Formlabs GmbH, Berlin, Germany) from the high-performance polymer Permanent Crown Resin A2 (PCR, Formlabs GmbH, Berlin, Germany). The PCR meets the conditions for intraoral retention according to the Medical Devices Regulation Class IIa [25]. The CAD software platform Fusion 360^®^ (Autodesk GmbH, Munich, Germany) was used to design the bracket geometry digitally, particularly the base design [26]. Two different retention patterns were chosen to investigate the influence of the base design (plain surface in group A = micro-retention; horizontal deepening bars in group B = macro-retention). Groups A and B were scaled to 9.67 mm^2^, corresponding to the discovery^®^ control group (group C = metal base). To investigate the effect of increasing the base area, another in-office bracket with a plain base and an area of 12.28 mm^2^ was fabricated (group D). The cross-sectional area of 12.28 mm^2^ is based on the shear bond strength of control group C. The objective was to scale the value for F_max_ to the value of control group C by increasing the cross-sectional area (Table 2).

Blender^®^ CAD software version 2.93.1(Blender Foundation, Amsterdam, The Netherlands) was used to customize the bracket base with an adaptation to the anatomy of the corresponding tooth surfaces. For this purpose, the bovine teeth were digitally impressioned using a surface scan (TRIOS 4, 3Shape A/S, Copenhagen, Denmark). In Blender^®^ software, the digitally designed brackets were positioned on the tooth surfaces and individually adapted to the tooth surface by subtraction (Figure 1). Subsequently, an individualized positioning aid was designed at the incisal edge. On the virtual printer platform PreForm (Formlabs GmbH, Berlin, Germany), the individualized brackets with positioning aids were set up and prepared for 3D printing with PCR.

After completion of the printing process, manual post-processing was carried out according to the manufacturer’s instructions. The post-processing procedure included in the first step the removal of excessive resin with a three-minute cleaning in a 99% isopropyl alcohol suspension in the Form Wash (Formlabs GmbH, Berlin, Germany). The second step was air-drying for 30 min. In the third step, the brackets were examined for resin residues, which were removed through selective post-cleaning with the alcohol suspension followed by air-drying. In the fourth step, the brackets were hardened in the Form Cure (Formlabs GmbH, Berlin, Germany) for 20 min at 60 °C and under ultraviolet light with a wavelength of 405 nm. In the fifth step, the support structures were removed, and the brackets were roughly finalized and cured again for 20 min at 60 °C. The sixth and final step was polishing with a pumice stone and polishing paste. The residues were removed with distilled water. For visual inspection, all brackets were examined for damage in the Keyence^®^ VHX500 digital microscope (Amray 1610T, Bedford, MA, USA) prior to testing (Figure 2).

Shear bond strength tests: The shear bond strength was measured according to DIN 13990 for “Dentistry—Test methods for shear bond strength of adhesives for orthodontic attachments” [23]. As a preliminary measure, the bovine teeth were embedded in a polymer (Technovit^®^ 4004, Kulzer GmbH, Wehrheim, Germany). The vestibular surface faced upward and was kept clear and cleaned for subsequent bracket placement. For this purpose, a pumice powder/water mixture of 50 g to 40 mL was applied to a linen brush (diameter 100 mm), and the teeth were cleaned. Cleaning was performed in the occlusal-gingival direction for three seconds and a rotation of 3000 min^−1^. Then the pumice mixture was again applied to the enamel and cleaning performed in the mesial-distal direction for three seconds. The surface was then cleaned again for three seconds in the occlusal-gingival direction.

The cleaned teeth were then etched with 35% phosphoric acid (Unitek™ Etching Gel REF 712-039, 3M Unitek, Saint Paul, MN, USA) for 30 s. The phosphoric acid was rinsed off with water, and the teeth were dried with air. The etched pattern was then visually inspected. A thin layer of Transbond™ XT Primer (3M Unitek, Saint Paul, MN, USA) was applied to the etched areas. At the same time, the bracket to be bonded was prepared. A small amount of Transbond XT™ Adhesive Paste (3M Unitek, Saint Paul, MN, USA) was applied to the bracket base. Transbond™ XT is a methacrylate-based adhesive consisting mainly of Bis-GMA and TEGDMA. Next, the prepared bracket was attached to the pretreated bovine teeth using positioning aids. The metal brackets were positioned freehand. The excess composite was removed, and the adhesive was exposed to the polymerization lamp (Hilux Ledmax 1055, Benlioglu Dental Inc., Ankara, Turkey) for 10 s each from distal and mesial at an angle of about 30 degrees and a distance of about 5 mm. The polymerization lamp met the requirements of DIN EN ISO 10650 [27]. The prepared samples were stored at 37 °C for 24 ± 2 h in a water bath with water quality three according to DIN ISO 3696 [24].

The embedded teeth were placed in a Zwick/Roell universal testing machine (type BZ1-MM14450.ZW03, ZwickRoell GmbH & Co. KG, Ulm, Germany). Particular care was taken to ensure that the bracket to be tested was set parallel to the applied shear force from the occlusal to the gingival surface. A shear blade was placed over the locked and adjusted samples. The blade was positioned precisely parallel to the lower edge of the bracket base (Figure 3).

In a further step, the shear blade was locked into the testing machine. Shear was then applied at a constant traverse speed of 1 mm/min until failure of the test bond. Simultaneously with the shear bond strength test, a force-displacement diagram was recorded for later evaluation. This recording also made it possible to detect any distortions due to possible sliding of the plate. The shear bond strength test was performed at 23 ± 2 °C. A total of 10 shear bond strength tests per test group (*N* = 10) were performed and recorded. All sheared brackets and the corresponding teeth were examined under a digital microscope (Keyence^®^ VHX500, Amray 1610T, Bedford, MA, USA) at ×20 magnification for adhesive residue and enamel damage (Figure 4).

Statistical analysis was performed using IBM SPSS statistical software version 28.0.1.0 for Windows (IBM, Armonk, NY, USA). For this purpose, the nonparametric Kruskall–Wallis test followed by the Dunn–Bonferroni post hoc test and Mann–Whitney U test were performed. The four groups were analyzed for significant differences in the maximum applied shear force (F_max_) and shear bond strength (force per area). The significance level was set at α = 0.05 with *p* < 0.05.

## 3. Results

The first series of tests included the shear of bracket bases A, B, and C. The highest shear bond strength values and maximum shear forces were observed for the metal brackets (group C) (SBS: 12.0 ± 3.8 MPa, F_max_: 115.7 ± 36.6 N). For the printed brackets, there was a significant difference in SBS and F_max_ between group A and group B (A: SBS 8.8 ± 2.3 MPa, F_max_ 84.7 ± 21.82 N; B: SBS 12.0 ± 2.1 MPa, F_max_ 106.5 ± 20.7 N). The brackets with base D were investigated in a further test run in the same test design. A significant difference in F_max_ was found between groups A and D (D: F_max_ 118.5 ± 22.8 N). A significant difference between bases B, C, and D for SBS and F_max_ could not be detected (Figure 5 and Figure 6). The ARI score was between 2 and 3 for base A, while it was around 1 for bases B, C, and D (Table 3).

## 4. Discussion

One of the most important aspects of the clinical performance of a novel orthodontic bracket is the shear bond strength. This should be tested extensively before introduction to the market. Ideally, it should be achieved in a standardized setup that has already served for the introduction of new bracket systems and allows comparability with them. A commercially available bracket that has been thoroughly investigated in previous studies should serve as the gold standard. In the presented in vitro study, the shear bond strength of 3D-printed brackets was compared with that of conventional metal brackets. The results are intended to orient the design of in-office manufactured bracket systems to ensure a sufficient adhesive bond between tooth and bracket for clinical use.

A standardized and established test method was used for the measurement. This test method, summarized as DIN 13990, allows good comparability due to the high degree of standardization and is the method used in numerous publications [18,28]. Currently this national standard has not yet been worked out to an international standard. It is to be noted that relevant aspects of the DIN 13990 were formulated based on the ISO standard 29022. However, a novel test method must also be mentioned at this point, in which printed plastic teeth act as the test object [29,30]. These probably have a lower fault tolerance than bovine teeth. However, it remains to be clarified which is more appropriate to the clinical situation. Moreover, it is still an open question whether an artificial tooth behaves similarly in artificial aging to a bovine tooth.

The aim of this study was to gain a first impression of a possible basic design and durability of 3D-printed individualized brackets compared with established conventional metal brackets. A simple design is particularly relevant for in-house production to achieve maximum efficiency with minimal effort. At this point, the study does not claim to provide fundamental insights into structure–property relationships. If this had been the claim, further tests would have to be carried out, which were not considered in the context of this study [31]. However, the tests carried out provide initial indications of fundamental aspects that should enable the clinical use of 3D-printed brackets in the future.

To calculate the shear bond strength (force per area), the base size (length × width) of the investigated brackets was used as the area. The base size of the printed brackets corresponded to the surface size (length × width) of the control group (discovery^®^). However, it must be mentioned at this point that the possible surface enlargement due to roughness, laser structuring, and macro-retentions in the sense of bars, as used in group B, were not considered. Surface roughness also influences the bonded connection, and the pattern of bars increases the total surface area, which may disturb the relationship of the bonded areas [32,33,34]. Thus, only an approximation in length and width to the surface size was made in the present study. Therefore, larger tolerance values must be expected.

Transbond™ XT was used as the adhesive. This is a light-curing adhesive used to attach brackets to the tooth surface. The chemical composition consists of several components. One component is bisphenol A glycidyl methacrylate (Bis-GMA). This is a resin that contributes to polymerization and has high adhesion strength. Another ingredient is triethylene glycol dimethacrylate (TEGDMA). This is another resin that reduces the viscosity of the adhesive and accelerates curing. Silica is also added, which is a filler that gives the adhesive its consistency and improves the bonding properties. Another filler is barium glass. The purpose of this is to give the adhesive a higher density to increase X-ray visibility. Camphorquinone is the initiator responsible for the polymerization of the adhesive by the action of light energy. Ethoxybenzoic acid serves as a stabilizer that increases the shelf life of the adhesive. Overall, it can be stated that Transbond™ XT is a methacrylate-based adhesive consisting mainly of Bis-GMA and TEGDMA. The adhesive has been investigated for its properties in various studies [35,36,37]. It creates a mechanical bond between the bracket and the adhesive. A chemical bond is created only between the adhesive and the bonding agent Transbond™ XT Primer. The bonding agent penetrates in advance into the surface structure of the enamel, which has been roughened by acid, and creates a mechanical bond. An increase in the bonding surface can thus lead to an improved adhesive bond. On the other hand, there are approaches to optimize the bracket bases with a micro-retentive design by individualizing the tooth surface and increasing the surface area for the adhesive bond. Customized lingual bracket systems should be mentioned here [38,39]. However, it can be assumed that a rough surface that is individually adapted to the tooth has a higher adhesion than a standard bracket, where the adhesive gap can vary considerably.

The transferability of these results to the clinical situation must be discussed. The main difference from the clinical situation here is the selection of the bovine teeth tested, the constant shear load, and the simulated aging. In the clinical situation, the load is neither associated with a continuous force nor with always the same direction of the force. Furthermore, the influence of the arch as a possible stabilizing element is missing [40].

The measured shear bond strength values of the metal brackets align with the results of previous studies [20,41,42,43,44,45]. The 3D-printed brackets with a macro-retentive base design or with an increase in the base cross-section achieve comparable values to the metal brackets in terms of maximum shear force F_max_. This result was not expected. Zielinski et al. and Liu et al. showed that polymer brackets achieve significantly lower measured values for shear bond strength and maximum shear force, respectively, compared with metal and ceramic brackets [11,12,17,30]. One reason for this could be that the printed brackets have an additional adaptation to the corresponding tooth surface in addition to the retentive design. Conventional bracket systems do not have an individualized base, which can lead to significant discrepancies between the bracket and tooth surface. The resulting more significant bonding gap can potentially lead to lower shear bond strength. Furthermore, a corresponding mismatch between the tooth surface and the bracket base can lead to first-order fit errors and thus reduce the efficiency of a bracket system. Another reason is probably the increased surface area created by the macro-retentive design.

The micro-retentive bases could reach the values of the metal brackets only to a limited extent. However, there is no significant difference between base A and base C. This is mainly due to the high standard deviation of the metal brackets. These deviations could again reflect a lack of individualization of the base to the tooth surface. The results of the macro-retentive and micro-retentive bases differ significantly, so it can be assumed that a macro-retentive base is more suitable for everyday clinical use than a micro-retentive base. Even with a micro-retentive design, enlarging the base can increase the maximum shear force. Whether enlarging the base in a tooth-colored bracket system leads to a significant deterioration of the esthetic appearance is still to be discussed and cannot be proven. However, the enlargement of the base could be an advantage from a biomechanical point of view. A larger base can be expected to lead to a more effective rotational movement [41].

In this study, it was shown that a macro-retentive design is favorable compared with a micro-retentive design with regard to the bond. The macro-retentive design could be further enhanced by increasing the number of undercuts.

Nevertheless, a possible positive influence on the bond strength of the micro-retentive design via sandblasting or chemical conditioning should be clarified in a further investigation. The advantage of a micro-retentive design can be seen in the possibility of a flatter bracket design. A flat and fully individualized bracket geometry can result in a more efficient fixed treatment compared with a conventional bracket geometry based on standard values. The flat bracket design moves the force application closer to the center of resistance, and the individualization allows precise first-, second-, and third-order movements to be implemented [46]. More efficient therapy results in a reduction in the known side effects of fixed therapy such as demineralization [47,48].

In addition, the micro-retentive base design is particularly suitable for sensitive enamel surfaces. The ARI in this group is between 2 and 3, which means that the adhesive adheres entirely or mainly to the enamel, and enamel cracks are, therefore, unlikely. The ARI of groups B, C, and D were between 1 and 2. These values indicate that parts of the adhesive adhere to the enamel surface and the bracket base. The lower the ARI value, the more likely that the enamel is damaged. The ARI values obtained are comparable to those of previous studies. Enamel damage was not observed in this study. However, this also occurs mainly with ceramic brackets. Previous studies have shown that brackets made of a resin material provoke enamel cracks less frequently than other materials [12,17,45]. This was confirmed in the present study. The approaches to in-office manufacturing of fully individualized bracket systems are promising but require further material science investigations and clinical studies.

## 5. Conclusions

Conventional metal brackets show the highest shear bond strength values in a standardized in vitro test procedure. Low loss rates and, consequently, fewer unscheduled appointments in clinical practice can be expected.Innovative 3D-printed bracket systems can meet the high clinical requirements if a base individualized to the tooth surface has a macro-retentive pattern.Scaling the cross-sectional area to model the bond strength between the tooth and bracket provides a further degree of individualization, in addition to color, prescription, and geometry, and can represent a further step towards the personalization of fixed orthodontic therapy.

## Figures and Tables

**Figure 1 jfb-14-00289-f001:**
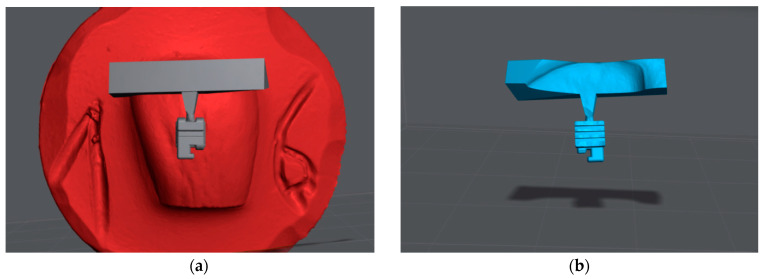
CAD procedure of the individualized bracket bases: (**a**) Positioning of the bracket with the positioning aid on the tooth surface; (**b**) Subtraction procedure for individualizing the bracket base with a perfect fit on the tooth surface.

**Figure 2 jfb-14-00289-f002:**
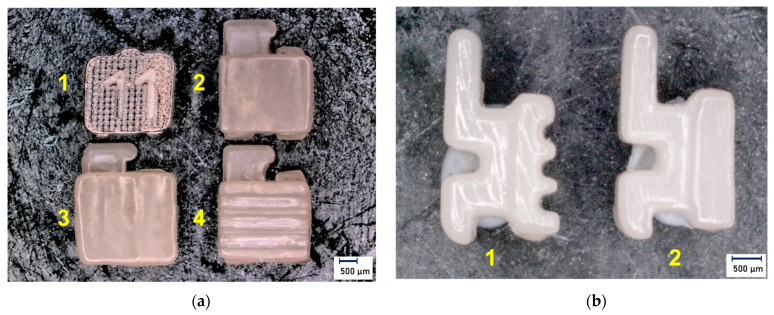
Imaging of the individualized bracket bases and the metallic control group (magnification ×20): (**a**) Overview of the different bracket bases (1—laser-structured base C discovery^®^ Dentaurum, 2—micro-retentive printed base A in-office, 3—enlarged micro-retentive printed base D in-office, 4—macro-retentive printed base B in-office); (**b**) Side view of the macro-retentive (1) and micro-retentive (2) design of the in-office bracket systems; light microscopic images.

**Figure 3 jfb-14-00289-f003:**
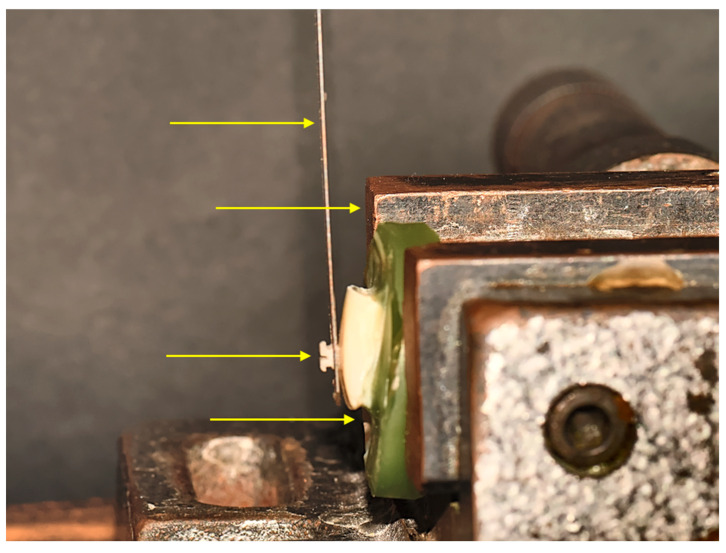
Test sequence in the Zwick/Roell universal testing machine. Clamped sample in the universal testing machine with correctly applied shear blade. The yellow arrows point from top to bottom to the following components: shear blade, Zwick/Roell machine, in-office manufactured bracket bonded to bovine tooth, prepared sample.

**Figure 4 jfb-14-00289-f004:**
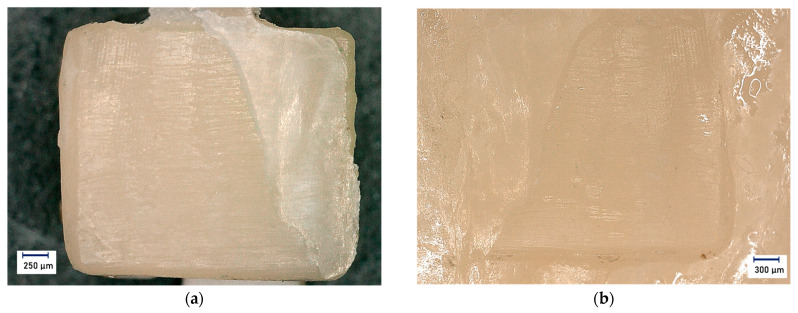
Representation of the ARI score (adhesive remnant index) 2 for a base A (magnification ×20): (**a**) Top view of the bracket base (base A) with approx. 1/3 adhesive residue; (**b**) Top view of the enamel surface with approx. 2/3 adhesive residue. Light microscope images.

**Figure 5 jfb-14-00289-f005:**
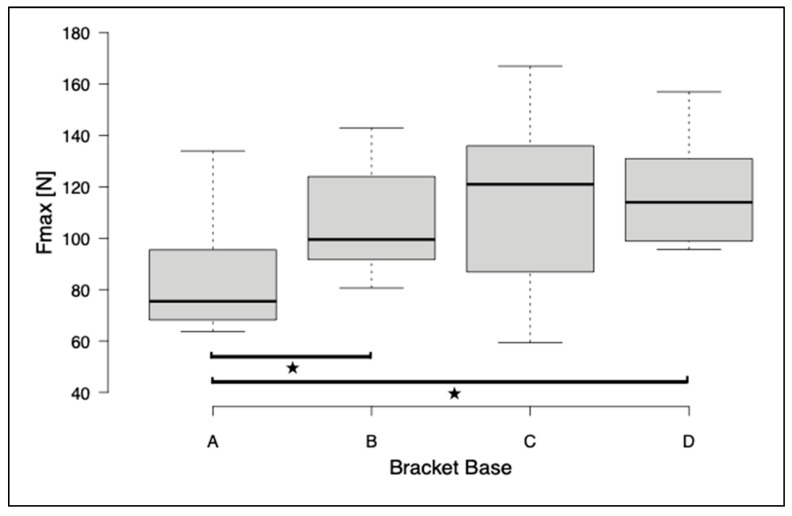
Box and whisker plot of the maximum force for all bracket bases. To determine significant differences, the Kruskal–Wallis Test and post hoc Dunn–Bonferroni tests were performed. The asterisks indicate significant differences (*p* ≤ 0.05).

**Figure 6 jfb-14-00289-f006:**
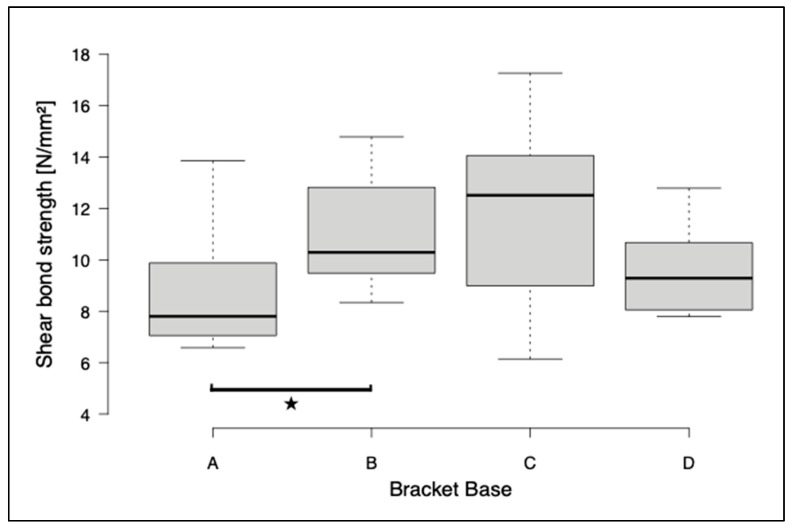
Box and whisker plot of the shear bond strength for all bracket bases. To determine significant differences, the Kruskal–Wallis test and post hoc Dunn–Bonferroni tests were performed. The asterisks indicate significant differences (*p* ≤ 0.05).

**Table 1 jfb-14-00289-t001:** Adhesive remnant index (ARI) and definition of the scores [19].

Score	Definition
0	No adhesive remained on the enamel.
1	Less than 50% of adhesive remained on the enamel.
2	More than 50% of adhesive remained on the enamel.
3	All adhesive remained on the enamel.

**Table 2 jfb-14-00289-t002:** Different types of brackets, manufacturer’s specifications, and mechanical properties of the tested materials.

Group	Bracket Type	Material	Base Morphology	Base Retention	Base Size (In mm^2^)
A	In-office printed	PCR	Individual	Micro-retention	9.67
B	In-office printed	PCR	Individual	Macro-retention	9.67
C	Discovery^®^	Metal	Standardized	Patented laser structured	9.67
D	In-office printed	PCR	Individual	Micro-retention	12.28

**Table 3 jfb-14-00289-t003:** Distribution frequency of the values of the ARI score.

Groups	*N*	ARI = 0 (%)	ARI = 1 (%)	ARI = 2 (%)	ARI = 3 (%)
Base A	10	0 (0)	0 (0)	4 (40)	6 (60)
Base B	10	0 (0)	9 (90)	0 (0)	1 (10)
Base C	10	1 (10)	8 (80)	0 (0)	1 (10)
Base D	10	0 (0)	7 (70)	2 (20)	1 (10)

## Data Availability

The data presented in this study are available on request from the corresponding author.

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
