# Peer review of "Influence of Individual Bracket Base Design on the Shear Bond Strength of In-Office 3D Printed Brackets—An In Vitro Study"

_jfb, 2023, doi:10.3390/jfb14060289_

Round 1
Reviewer 1 Report
The manuscript describes the evaluation of the different design options of the in-office 3D printed bracket base in terms of adhesive bond between the bracket and tooth, measured as shear bond strength (SBS) and maximum force (Fmax).
The topic is novel and of practical significance since 3D printing is also used in dentistry. However, the methods are overly simple, in only one test (shear bond strength test) the output is strength which is calculated as force/surface area. Using force as a separate result is far stretched (Fig. 5 and 6 having the same trend and essentially the same result). There are simply not sufficient parameters to quantify any practical differences and draw any valid conclusions.
Morphology is in the title, but it was not investigated. Also, since there are grooves, the base area does not equal to contact/interfacial area. Surface roughness which is related to mechanical retention has not been investigated nor discussed.
The discussion is not thorough. What is the composition of the adhesive? Is there chemical bonding between printed brackets (resin) and adhesive? And is this bonding different from the bonding between the metal and the adhesive? These have not been considered nor discussed.
The first sentence in Introduction is not complete.
Thus a major revision is necessary to remedy these before further consideration.
Some suggestions:
1. Fig. 2, 4, can you add the scale bars to put the size in perspective.
2. Fig. 3 has large black background. Please reduce it and add pointers to the main components.
3. “A total of 10 shear bond strength tests were performed and recorded for four test groups,” how many tests for each group?
4. Both 20x and x20 have been used. Please be consistent. Use proper multiply sign rather than letter x.
5. What alfa value was used for the statistical analysis?
6. Fig. 5, 6, left axis label should be written in the format parameter / unit, Force / N and Shear bod strength / N mm-2
Base letters should be in capital letters.
7. Since A-D have been defined, there is no need to repeat many times later on in figure captions and table titles.
8. It may be necessary to present the calculation of strength, strength=force/area
9. Fig. 2, the ordering CADB is odd.
10. It is recommended to cite doi:10.3390/met12030406 when discussing surface characteristics and surface area in relation to retention and performance.
it is fine.
Author Response
Dear Reviewer,
Thank you for your effort and time to prepare a review for our study. We took your comments and suggestions for improvement very seriously and worked through them point by point.
The topic is novel and of practical significance since 3D printing is also used in dentistry. However, the methods are overly simple, in only one test (shear bond strength test) the output is strength which is calculated as force/surface area. Using force as a separate result is far stretched (Fig. 5 and 6 having the same trend and essentially the same result). There are simply not sufficient parameters to quantify any practical differences and draw any valid conclusions.
- We can understand the point of criticism. The method is kept simple in its structure. However, we strictly followed the test procedure according DIN Norm. This represents an established and standardized procedure that allows good comparability to other bracket systems that have undergone such a test procedure. Bond strength is one of the most important aspects for the clinical performance of an orthodontic bracket and should be tested prior to bringing it onto the market, best in a standardized set-up. A commercially available bracket that has been studied extensively in earlier studies should serve as a gold standard. We thus decided to apply the testing method described in the DIN 13990 and to use the bracket discovery as a reference.
Morphology is in the title, but it was not investigated. Also, since there are grooves, the base area does not equal to contact/interfacial area. Surface roughness which is related to mechanical retention has not been investigated nor discussed.
- The surface roughness certainly has an effect on the adhesive bond, and the groove structure also increases the overall surface area, which disturbs the bond area ratio. However, we also followed the ISO standard here and just spread the dimension of the base of the control group with length times width. However, this point is certainly to be taken into account and therefore we have subsequently included it in the discussion. Here we also made it clear that the exact surface size was not determined, but only an approximation in length and width took place. We must therefore assume larger tolerance values.
The discussion is not thorough. What is the composition of the adhesive? Is there chemical bonding between printed brackets (resin) and adhesive? And is this bonding different from the bonding between the metal and the adhesive? These have not been considered nor discussed.
- We have added the composition of the adhesive. There is no chemical bond between printed brackets and adhesive because the resin used is not a polycarbonate. We added it to discussion.
The first sentence in Introduction is not complete.
- The first sentence has of course been corrected, please excuse this clerical error.
Thus a major revision is necessary to remedy these before further consideration.
Some suggestions:
- Fig. 2, 4, can you add the scale bars to put the size in perspective.
- Fig. 2 and 4 have been provided with scale bars - Fig. 3 has large black background. Please reduce it and add pointers to the main components.
- The background of Fig. 3 was lightened, and the individual components were provided with arrows. The description is now in the caption. - “A total of 10 shear bond strength tests were performed and recorded for four test groups,” how many tests for each group?
- 10 brackets per group have been tested. The sentence has been corrected. - Both 20x and x20 have been used. Please be consistent. Use proper multiply sign rather than letter x.
- Multiply sign was used and x20 was given consistently. - What alfa value was used for the statistical analysis?
- The alfa value was added. - Fig. 5, 6, left axis label should be written in the format parameter / unit, Force / N and Shear bod strength / N mm-2 . Base letters should be in capital letters.
- The labels of the axes of Fig. 5 and 6 have been designed according to your wishes. - Since A-D have been defined, there is no need to repeat many times later on in figure captions and table titles.
- The definitions were removed from the captions - It may be necessary to present the calculation of strength, strength=force/area
- Strength has been investigated as shear bond strength (force/area) - Fig. 2, the ordering CADB is odd.
- The arrangement in Fig. 2 is for illustration of the base sizes, since they are based on control group C. The order in Fig. 2 has been changed. For this reason, the order has been left as it is. If you have a compelling need for a change, we would exchange the group labels altogether. - It is recommended to cite doi:10.3390/met12030406 when discussing surface characteristics and surface area in relation to retention and performance.
- The study has been included.
Thank you for the great suggestions for improvement and the comprehensive review, we have tried to implement all suggestions.
With kind regards
the authors
Reviewer 2 Report
Thank you for submitting this paper and allowing me to review this:
I do have some suggestions to make
1. The authors write
Introduction: approaches demonstrate modern CAD/CAM solutions for in-office manufacturing 42 of fully customizable bracket systems made of high-performance polymers. Is this part of an incomplete sentence. Please correct.
2. There are several limitations of using SBS.
The flat surface of the tested specimen does not resemble the normally contoured tooth-bonded surface. The study settings, particularly the SBS test, does not reflect the clinical setting for removing orthodontic brackets, which may affect the failure mode. In addition, the oral cavity presents a different testing environment regarding temperature changes, saliva, and pH levels, which may also affect the outcomes.
Reference: Biadsee A, Rosner O, Khalil C, Atanasova V, Blushtein J, Levartovsky S. Comparative evaluation of shear bond strength of orthodontic brackets bonded to three-dimensionally-printed and milled materials after surface treatment and artificial aging. Korean J Orthod. 2023 Jan 25;53(1):45-53.
3. Would have been very beneficial to demonstrate the SEM images for good visualization.
4. Any comments on cohesive failure of 3D printed brackets?
5. Did you consider measuring the surface free energy or other micro-shear bond tests as macro and micro tests often reveal different values.
Minor grammatical corrections required
Author Response
Dear Reviewer,
Thank you for your time and effort in preparing a review for our study. We have taken your comments and suggestions for improvement very seriously and have worked through them point by point.
- The authors write
Introduction: approaches demonstrate modern CAD/CAM solutions for in-office manufacturing 42 of fully customizable bracket systems made of high-performance polymers. Is this part of an incomplete sentence. Please correct.
- The first sentence has of course been corrected, please excuse this clerical error.
- There are several limitations of using SBS.
The flat surface of the tested specimen does not resemble the normally contoured tooth-bonded surface. The study settings, particularly the SBS test, does not reflect the clinical setting for removing orthodontic brackets, which may affect the failure mode. In addition, the oral cavity presents a different testing environment regarding temperature changes, saliva, and pH levels, which may also affect the outcomes.
- We can understand the point of criticism. The method is kept simple in its structure. However, we strictly followed the test procedure according to DIN-Norm 13990. This represents an established and standardized procedure that allows good comparability to other bracket systems that have undergone such a test procedure. We know that the results are not comparable to the in vivo situation. For this reason, we have tried to clearly describe in the discussion that this in vitro study only provides an indication for possible clinical use and that further in vivo studies are needed. Bond strength is one of the most important aspects for the clinical performance of an orthodontic bracket and should be tested prior to bringing it onto the market, best in a standardized set-up. A commercially available bracket that has been studied extensively in earlier studies should serve as a gold standard. We thus decided to apply the testing method described in the DIN 13990 and to use the bracket discovery as a reference.
Reference: Biadsee A, Rosner O, Khalil C, Atanasova V, Blushtein J, Levartovsky S. Comparative evaluation of shear bond strength of orthodontic brackets bonded to three-dimensionally-printed and milled materials after surface treatment and artificial aging. Korean J Orthod. 2023 Jan 25;53(1):45-53.
- Interesting work, which is also known. Since the teeth are acrylic, the margin of error is probably smaller. But it still needs to be clarified which is more in line with the clinical situation. In addition, it is still an open question whether an artificial tooth behaves similarly to a bovine tooth during artificial aging.
- Would have been very beneficial to demonstrate the SEM images for good visualization.
- If a lot of acrylic remains on the bracket, then enamel cracks are to be expected. If most of the composite remains on the tooth, enamel avulsion cannot occur. For this reason, the ARI score is used to make an initial assessment of whether or not there is an increased risk of enamel avulsion. For this reason, no SEM images were taken. - Any comments on cohesive failure of 3D printed brackets?
- Thank you for pointing this out. We have included the point in the discussion. - Did you consider measuring the surface free energy or other micro-shear bond tests as macro and micro tests often reveal different values.
- Other microshear bond tests we have considered. However, for the first step, we wanted to stick to an established standardized protocol that has been used for launching novel bracket systems. This allows us to get a first impression of the shear bond strength and on the basis of which further refinements are possible.
Thank you for the great suggestions for improvement and the comprehensive review, we have tried to implement all suggestions.
With kind regards
the authors
Round 2
Reviewer 1 Report
The authors have addressed all issues raised in the revision.
Minor points to address:
1. please refer to the international (ISO) standards before any local ones, in international peer-reviewed journals such as JFB.
2. The scale bars in Fig. 2, 4 are disproportionally small, please adjust.
Author Response
Dear Reviewer,
Thank you very much for the quick feedback and the interesting comment. At present, this national standard has not yet been developed into an international standard. However, relevant aspects of DIN 13990 have been formulated in accordance with the ISO 29022 standard. The ISO 29022 standard is also currently being revised in ISO TC105. However, it has not yet been finalized. We have attached this point to the discussion. Since the procedure according to DIN 13990 has nevertheless also been published in international journals, we think that the manuscript can be a scientific contribution to the future use of in-office printed brackets.
Furthermore, the scale bars have been enlarged.
With kind regards, the authors.